# Research on the Action and Mechanism of Pharmacological Components of *Omphalia lapidescens*

**DOI:** 10.3390/ijms252011016

**Published:** 2024-10-13

**Authors:** Keyang Xu, Li Wang, Dan He

**Affiliations:** Department of Pathogenobiology, Key Laboratory of Pathobiology, Ministry of Education, College of Basic Medical Sciences, Jilin University, Changchun 130021, China; xuky23@mails.jlu.edu.cn (K.X.); wli99@jlu.edu.cn (L.W.)

**Keywords:** *Omphalia lapidescens*, pharmacological ingredients, medicinal fungus, antineoplastic

## Abstract

*Omphalia lapidescens* is a macrofungus that is used in traditional Chinese medicine for its insecticidal and stagnation-relieving properties. The active ingredients of this fungus including proteins, polysaccharides and sterols have been demonstrated to exhibit antiparasitic, anti-inflammatory, and antitumor effects. *Omphalia* has been used in clinical cancer treatment. Many studies on *Omphalia* have concentrated on its cytotoxicity and anticancer effects. However, the investigation of its natural metabolites remains a significant area for further research. This review presents a comprehensive analysis of the research progress concerning the pharmacological components of *Omphalia*. The aim of this discussion is to provide a reference for further in-depth study of *Omphalia*, with the objective of exploring its potential value. Therefore, the focus of this review was on the classification of metabolites in *Omphalia* and their mechanisms of action.

## 1. Introduction

*Omphalia lapidescens* has long been regarded as a crucial antiparasitic herb in traditional Chinese medicine, whose importance was particularly pronounced when parasitic diseases were prevalent across China. The initial national survey on parasitic infections published in 2005 indicated that over 50% of the Chinese population was infected with parasitic organisms, with infection rates exceeding 70% in children [1]. The third national survey on major human parasitic diseases published in 2019 indicated a notable decline in the overall infection rate, which had dropped to below 6% [2,3,4,5,6,7]. In addition to the efforts of medical personnel and the government, the listing and production of *O. lapidescens*-related drugs also played a positive role during this period. However, the southeastern coastal regions and northeastern provinces of China remain endemic areas for *Clonorchis sinensis*, and the seropositivity rate for *Toxoplasma gondii* antibodies in newborns in Fujian Province is as high as 9.38% [8,9]. However, the diagnosis and treatment of parasitic diseases remain challenging in clinical practice.

Several antiparasitic drugs including tribendimidine, praziquantel, and artemisinin are effective in treating parasitic infections and are commonly used in clinical practice [10,11]. However, these drugs are associated with toxic side effects on the heart, digestive system, liver, and kidneys. Furthermore, the prolonged use of these drugs may result in the emergence of drug resistance [12,13,14]. Research has thus focused on the broad-spectrum efficacy and safety of traditional antiparasitic herbs.

*O. lapidescens*, commonly known as “Lei Wan” or “Zhu Ling,” is a medicinal fungus that is widely distributed in western China. *O. lapidescens* forms sclerotia underground, much like black truffles, setting it apart from ordinary edible fungi. The sclerotia can be found on the roots of bamboo during the summer and autumn season; after winter, new fruiting bodies are produced by sclerotium, which are generally not easy to see [15].

*O. lapidescens* was historically linked to the large intestine and stomach meridians and used for the treatment of parasitic infestations and the promotion of digestion. Despite its long history of use, there has been a debate regarding the classification and scientific designation of *O. lapidescens*. The Encyclopedia of Chinese Materia Medica, the Pharmacopoeia of the People’s Republic of China, and the Complete Collection of Chinese Medicine frequently categorize *O. lapidescens* as a member of the *Tricholomataceae* family, with the scientific name *Omphalia lapidescens* [16]. However, the 2020 publication Chinese Medicinal Fungi reclassified it under the genus *Laccocephalum*, with the scientific name *Laccocephalum mylittae* [17]. Recent phylogenetic analyses based on fruiting body morphology and internal transcribed spacer sequencing have indicated that *O. lapidescens* is more closely related to the family Marasmiaceae and the genus *Gerronema* [18].

Species classification and genomics are inseparable in modern biology. Clear species classification deepens our understanding of an organism’s genome, enabling more accurate localization of metabolic gene clusters. This is particularly crucial in fungal genomes, where identifying species helps to facilitate in-depth research into fungal metabolic pathways, the biological functions of their metabolites, and their adaptation to diverse ecological environments. By comparing metabolic gene clusters across species, it is possible to trace the evolutionary history of metabolic pathways and predict their future development. This understanding is essential for advancing drug discovery, synthetic biology, and improving artificial cultivation techniques. In conclusion, based on our predecessor’s research, it is recommended that *O. lapidescens* be classified under the genus *Gerronema lapidescens*.

Given the taxonomic complexity of *O. lapidescens lapidescens* (Lei Wan), comprehensive research into its metabolites and mechanisms of action is imperative. This review integrates studies on the pharmacological components of *O. lapidescens*, providing a detailed overview of the pharmacological effects, chemical constituents, and mechanisms of action. By integrating current knowledge, this review aims to offer valuable insights into potential research directions and the development of *O. lapidescens* components for medical applications.

## 2. Pharmacological Components of *O. lapidescens*

### 2.1. Protein

#### 2.1.1. Protease

Two distinct proteases have been isolated from *O. lapidescens*: a thiol protease and metalloprotease. The thiol protease isolated from *O. lapidescens* has a molecular weight of 16.8 kDa, constituting approximately 3% of the sclerotium. This protease exhibits the ability to hydrolyze common esters and demonstrates optimal activity at 37 °C and pH 7.48. Amino acid composition analysis indicates that the thiol protease contains approximately 7.25% neutral sugars, with a total sugar content of about 16.7%. It is particularly rich in methionine and acidic amino acids, with the methionine content reaching 31.5%, while the content of basic amino acids is relatively low. Furthermore, this protease displays proteolytic activity that is not inhibited by EDTA or other metal ions and shows hydrolytic activity toward esters, with weak hydrolytic activity against *M. Lysodeikticus* [19].

The metalloprotease isolated from *O. lapidescens* consists of subunits with a molecular weight of approximately 4 kDa, though its content in the sclerotium remains undefined. This enzyme possesses casein hydrolytic activity, exhibiting optimal hydrolytic performance at 50 °C and pH 7.5. Experimental evidence indicates that disulfide bonds within the metalloprotease are crucial for maintaining the molecular folding required for its activity. This enzyme is significantly activated by Ca^2+^, while other divalent metal ions, such as Cu^2+^, Co^2+^, and Mn^2+^, inhibit its activity. In contrast, monovalent ions such as K^+^ and Na^+^ do not significantly affect its activity. This observation suggests that Ca^2+^ may serve as the primary activator, playing a vital role in stabilizing its tertiary structure and protecting the enzyme from thermal denaturation and autohydrolysis [20].

#### 2.1.2. Lectin

The lectin isolated from *O. lapidescens* has a molecular weight of approximately 12 kDa and constitutes about 1% of the sclerotium. It is characterized as a single peptide chain, with no sugar components present. This lectin exhibits agglutination activity against rabbit and human blood. Ca^2+^, Mn^2+^, and Mg^2+^ do not significantly affect its activity, while Zn^2+^ enhances its specific activity fourfold [21].

#### 2.1.3. Protein pPeOp

Using polyvinylpyrrolidone (PVP) for extraction followed by molecular sieve chromatography, the proteins extracted from *O. lapidescens* were separated into three fractions. The fraction with the highest content was designated as pPeOp, with a molecular weight of approximately 16 kDa [22]. However, no related physicochemical properties have been investigated for this protein.

### 2.2. Polysaccharides

The study of bioactive polysaccharides has been a prominent area of research across a range of scientific disciplines globally for many years. Fungal polysaccharides have been demonstrated to possess a range of beneficial properties including antitumor, antiviral, immunomodulatory, antidiabetic, anticoagulant, and antioxidant effects [23,24,25,26,27]. In recent years, enoki mushroom polysaccharides have been shown to possess potent anti-inflammatory properties [28]. The recently discovered polysaccharides derived from *Ganoderma lucidum* have been shown to effectively mitigate intestinal ischemia-reperfusion injury [29] and promote diabetic wound healing [30]. Given the extensive bioactivity of fungal polysaccharides, research on *O. lapidescens* has also focused on the study of its polysaccharides. The content of polysaccharides in *O. lapidescens* is about 2.3% [31].

#### 2.2.1. Polysaccharide S4001 and S4002

Two polysaccharides, S-4001 and S-4002, with relative molecular masses were isolated from *O. lapidescens* by gel filtration. The chemical structure of S4001 is that of a dextran comprising a β(1–3) glucose main chain and (1–6) branched chains, with an average molecular weight of 1,183,000 [32].

#### 2.2.2. Polysaccharide OL-1, OL-2, and OL-3

Japanese researchers have isolated three polysaccharides from *O. lapidescens* species: OL-1, OL-2, and OL-3. The main component of OL-2 is β-(1→3)-D-glucan, which contributes to its more stable gel structure [33]. OL-3, a highly branched polysaccharide, is composed of D-glucose, 2-acetamido-2-deoxy-D-glucose, and D-glucuronic acid. Detailed structural analyses using methylation, partial acid hydrolysis, and mass spectrometry has identified several glucopyranosyl linkages: (1→3), (1→4), (1→6), and (1→3,6). Additionally, the following structures were identified: 2-acetamido-2-deoxy-β-D-glucopyranosyl-(1→4)-β-D-glucuronosyl aldol, β-D-glucopyranosyl-(1→4)-β-D-glucopyranosylaldehyde, and β-D-glucuronosyl-(1→4)-2-acetamido-2-deoxy-β-D-glucopyranosyl-(1→4)-2-deoxy-β-D-glucose. These fragments suggest a complex, branched structure for OL-3 [34].

### 2.3. Triterpenes

Triterpenoids represent a significant class of natural products with notable bioactivities, serving as pivotal active components in numerous traditional Chinese medicinal herbs. To date, over 20,000 triterpenoids have been identified, with over 400 having been developed into drugs. These include numerous important steroid hormone drugs and natural products such as ginsenosides, which are widely used clinically. The recent approval of the novel antifungal triterpenoid drug Brexafemme reinforces the significance of triterpenoids in the pharmaceutical field.

Triterpenoids comprise 30 carbon atoms, typically forming polycyclic structures and containing a variety of functional groups including hydroxyl, carbonyl, and ester groups.

Several triterpenoid compounds have been isolated from *O. lapidescens* including ganoderic acid, eburicoic acid, and lanostane derivatives.

#### 2.3.1. Tetranorlanostane Triterpenoid

There is a new tetranorlanostane triterpenoid isolated from *O. lapidescens,* named (20S)-3β-hydroxy-24,25,26,27-tetranorlanost-8-ene-21(23)-lactone, and whose structural formula is shown in Figure 1 [35]. It is a white amorphous powder with the molecular formula C_26_H_40_O_3_, indicating seven unsaturations. NMR data showed that this component has a hydroxyl-substituted lanost-8-ene skeleton similar to that of eburicoic acid.

#### 2.3.2. Eburicoic Acid and Ganoderma Side D

Eburicoic acid (Figure 2 compound **1**) is a triterpenoid compound and a key constituent of the *Ganoderma* genus or *Antrodia camphorata*. It features a tetracyclic triterpene backbone with a carboxyl group at the C-3 position as well as hydroxyl groups typically at the C-15 and C-7 positions [36].

Ganoderma side D (Figure 2 compound **2**), another triterpenoid compound derived from *Ganoderma* fungi, has a tricyclic triterpene structure, characterized by carboxyl and hydroxyl functional groups. It resembles other ganoderic acids, with a 4,4,8-trimethyltricyclo[5.3.1.0]dodecane skeleton and functional groups attached at various positions [37].

Both compounds exhibit high stability and potent biological activities, making them valuable in pharmacological research, particularly in studies related to cancer treatment and immune modulation.

### 2.4. Ergosterol

The initial identification of sterols in fungi was achieved through the examination of *Claviceps purpurea*, which subsequently led to the designation of ergosterol, a pivotal constituent of fungal cell membranes. Ergosterol is a steroidal compound with a typical tetracyclic steroidal skeleton (A, B, C, and D rings), which is common to all sterol molecules. Ergosterol has antibacterial [38], antioxidant [39], and cholesterol-lowering effects [40] and can be converted into vitamin D in the body by ultraviolet light. The precursor of vitamin D2 belongs to ergol, which can be converted into vitamin D2 after absorption by sunlight. Compounds of ergosterol are also precursors to anti-inflammatory and hormonal drugs [41,42]. Ergosterol plays a very important role in ensuring membrane fluidity and its structural integrity, cell viability, and material transport [43].

A variety of ergosterols have been extracted from *O. lapidescens*. Based on their function, these were classified in Table 1, and the new ergosterols of *O. lapidescens* are listed in Figure 3.

### 2.5. Other Components

The petroleum ether extract of *O. lapidescens* sclerotia contained a variety of volatile oil components. Of these, palmitic acid, ethyl palmitate, and dibutyl phthalate were found in relatively high concentrations, accounting for approximately 25% of the volatile components. These three compounds are commonly used in the manufacture of insecticides in the industrial and agricultural sectors, and their combined effects may contribute to the insecticidal activities observed with *O. lapidescens* [55].

## 3. Biological Activity

### 3.1. Antiparasitic

As documented in the *Compendium of Materia Medica*, *O. lapidescens* is used to treat infections by Taenia, a genus of tapeworms.

The efficacy of *O. lapidescens* treatment was better than that of traditional anthelmintic drugs such as albendazole and comparable to that of praziquantel [56]. Furthermore, *O. lapidescens* showed good anthelmintic effects in studies involving tapeworms, *Ascaris*, *Spirometra mansoni*, and *Giardia lamblia* [57,58].

Electron microscopy demonstrated that parasitic tissues treated with *O. lapidescens* proteases exhibited substantial damage, characterized by extensive necrosis in the cortical and parenchymal regions of the parasite [59,60].

### 3.2. Antitumor

The antitumor activities of fungal ergosterol have been demonstrated in functional studies. Extracts of sterols and fatty acids derived from *Flammulina velutipes* potently inhibit the proliferation of human liver cancer cells (HepG2), glioma cells (U251), gastric adenocarcinoma cells (SGC), and lung adenocarcinoma cells (A549) [61,62,63]. Extracts of ergosterol derived from the spores of *Ganoderma applanatum* and *Ganoderma lucidum* have been shown to induce apoptosis in tumor cells through a number of different pathways [64,65].

Notably, in 1988, Yao et al. conducted research on the antitumor effects of *O. lapidescens* and found that *O. lapidescens* protein extracts inhibited the growth of ascites sarcoma 180 tumors in mice [66]. Tablets and capsules containing *O. lapidescens* as a principal component have since been extensively employed in clinical tumor treatments. The combination of *Omphalia* capsules with chemotherapy has been demonstrated to reduce the toxicity of chemotherapy drugs and stimulate hematopoiesis in the bone marrow, thereby enhancing the treatment effectiveness by 10% [67]. Furthermore, the combination of *Omphalia* capsules with gemcitabine for the treatment of advanced lung cancer has been demonstrated to improve patient survival rates and quality of life. In particular, the *Omphalia* capsule group exhibited a 32.6% enhancement in clinical efficacy and a 20% improvement in disease control [68].

The main components of *O. lapidescens* with antitumor properties include pPeOp protein, polysaccharides, terpenes, and ergosterol.

#### 3.2.1. *O. lapidescens* Protein pPeOp

In research on the mechanism of the antitumor effects of *O. lapidescens*, the proteases were extracted using a polyvinylpyrrolidone solution and purified by gel column chromatography, yielding a *O. lapidescens* protein designated pPeOp, with a molecular weight of approximately 16 kDa. pPeOp exhibited a concentration-dependent inhibition of the migration ability of MC-4 gastric cancer cells, without significant toxicity to normal gastric cells [22,69].

In both the MC-4 gastric cancer cells and SGC-7901 gastric adenocarcinoma cells, pPeOp significantly reduced the expression levels of key proteins in the JAK-STAT signaling pathway including JAK1, JAK2, and STAT3. It is noteworthy that the expression of SOCS3 and SOCS1, negative feedback regulators of this pathway, was upregulated. This led to the hypothesis that the tumor-suppressing mechanism of the pPeOp protein might be to block the proliferation of cancer cells through the upregulation of the negative feedback regulators and thus the negative regulation of the JAK/STAT3 signaling pathway. The JAK-STAT pathway plays a crucial role in cellular signaling and participates in processes such as cell proliferation, migration, and immune evasion [70]. The JAK-STAT pathway, which exhibits key functions in cell proliferation, migration, and immune evasion, is often activated in cancer [71,72]. These findings suggest that *O. lapidescens* pPeOp inhibits this pathway, thereby suppressing the migration and invasion of gastric cancer cells.

Further research on this pathway revealed that the *O. lapidescens* pPeOp protein arrests MC-4 and SGC-7901 gastric cancer cells in the S phase of the cell cycle while having no effect on the cell cycle of MC-1 normal gastric mucosal cells [73]. IL-6, an activator of the JAK-STAT pathway, combined with pPeOp amplified the inhibitory effect of *O. lapidescens* pPeOp on this pathway. The combined use of these agents significantly enhanced the cell cycle arrest effect on gastric cancer cells [74].

In recent years, studies on microRNAs have demonstrated their significant potential as biomarkers for early cancer screening. Following treatment of HGC-27 gastric cancer cells with pPeOp protein, a significant increase in the expression level of miR-30b-5p was observed compared with the levels in the untreated control group. This indicates that pPeOp may be involved in the upregulation of miR-30b-5p. Deep sequencing of miRNAs in HGC-27 cells and MC-1 normal gastric mucosal cells revealed a high expression of miR-30b-5p in cancer cells. Similarly, miR-30b-5p was expressed at significantly lower levels in clinical gastric cancer tissues compared with adjacent normal tissues. Transfection of miR-30b-5p mimics into cancer cells resulted in a significant inhibition of the migration, invasion, and proliferation of gastric cancer cells. The antitumor effect was significantly enhanced when the pPeOp protein was combined with other agents. The results of miRNA database analysis and dual-luciferase reporter assays confirmed that miR-30b-5p targets RAB22A. Western blot analysis and cytoskeleton fluorescence staining revealed that miR-30b-5p targeting RAB22A resulted in the downregulation of Rac1/Cdc42 expression and activation. This led to alterations in downstream pathways, which in turn disrupted the cytoskeletal structure and inhibited cancer cell proliferation, migration, and invasion [75]. We summarized the results of the existing published articles and outlined the antitumor mechanism of the protein pPeOp, which is shown in Figure 4.

#### 3.2.2. *O. lapidescens* Polysaccharides OL-2

OL-2, administered intraperitoneally and topically, had little or no antitumor activity against solid type 180 sarcoma but was effective against the ascites form of type 180 sarcoma. Interestingly, OL-2 also showed significant antitumor activity against the ascites form of MH-134 when administered with 5-fluorouracil, significantly prolonging the survival time of the mice (39 to 46 days). These results suggest that OL-2 has unique characteristics in terms of physicochemical properties and antitumor activity.

The Smith degradation of OL-2 using periodate oxidation yielded three degradation products: OL-2I, OL-2II, and OL-2III. Structural and physicochemical analyses indicated that the first two products, which exhibited a higher proportion of branched chains, demonstrated enhanced antitumor activity [33,76]. This finding lends support to the hypothesis that polysaccharides with a greater number of glucan branches possess elevated antitumor activity.

#### 3.2.3. Triterpene

Eburicoic acid exhibited significant cytotoxicity against both MDA-MB-231 breast cancer cells and HGC-27 gastric cancer cells. Their IC50 values were 29.63 ± 1.69 μM and 16.03 ± 1.30 μM, respectively [36].

#### 3.2.4. *O. lapidescens* Ergosterol

Lee et al. noted that introducing an epoxy group into the tetracyclic skeleton of ergosterol derivatives significantly enhanced their cytotoxic properties [77]. By isolating a series of 5α,6α-epoxides from the macrofungal *O. lapidescens*, they were able to establish key structure–activity relationships. The study highlighted the crucial role of double bonds in enhancing cytotoxicity, providing important insights into the potential of these compounds for cancer treatment [78]. In addition, there are many ergosterols that contain hydroxyl groups or polyols. A class of ergosterols with a ketone group was classified from *O. lapidescens*, among which ergosta-4,6,8(14),22-tetraen-3-one (Figure 5 Compound **1**) is commonly found in agarics. The interaction between this ergone and human serum albumin is mainly mediated by hydrogen bonds and hydrophobic interactions.

Fei Liu and colleagues isolated (22E,24R)-ergosta-7,9(11),22-trien-3β,5β,6α-triol (Figure 5 Compound **2**) from *O. lapidescens*, which exhibited potent cytotoxicity against human breast cancer cells [35]. Further research suggested that this cytotoxic effect may be associated with upregulation of the Bax/Bcl-2 ratio and downregulation of procaspase-3 expression, although the precise mechanism remains unknown. In a study conducted by Yue Wang et al., several ergosterols including (3β,5α,6β,22E)-6-methoxyergosta-7,22-dien-3,5-diol (Figure 5 Compound **3**) were isolated from *O. lapidescens* and showed excellent inhibitory activity against the HGC-27 gastric cancer cell line and non-small cell lung cancer cells. The antitumor activity of these compounds, particularly (3β,5α,6β,22E)-6-methoxyergosta-7,22-dien-3,5-diol, is thought to be mediated by the activation of caspase-3, which promotes apoptosis [79]. These findings further support the pharmacological potential of *O. lapidescens*-derived steroid compounds, which have both antibacterial and antitumor properties.

Caspase-3, a member of a family of cysteine proteases, is a key executioner protease in apoptosis (programmed cell death) [80]. It normally exists in an inactive form, called procaspase-3, and is activated in response to external signals such as DNA damage, drug exposure, or oxidative stress. Once activated, caspase-3 cleaves a variety of intracellular substrates including DNases, structural proteins, and signaling molecules, ultimately leading to apoptosis [81]. Because of its central role in apoptosis, caspase-3 is often referred to as the “executioner” of cell death. The fact that steroid compounds from *O. lapidescens* can activate caspase-3 suggests that these compounds have significant pro-apoptotic potential. The induction of apoptosis in cancer cells is a key mechanism of antitumor drugs. Therefore, the compounds found in *O. lapidescens* may provide a valuable foundation for the development of new anticancer drugs, particularly those that selectively target cancer cells by promoting the activation of caspase-3. Further research into the mechanisms of these steroid compounds could lead to more effective cancer treatments with reduced side effects compared with current chemotherapy options.

### 3.3. Anti-Inflammatory

The research on *O. lapidescens* polysaccharides dates back to the 1980s, when studies revealed that *O. lapidescens* polysaccharide S-4002 exhibited excellent anti-inflammatory and immune-enhancing properties in mice [32]. The proposed mechanism of action of S4002 is believed to be associated with corticosteroid activity.

### 3.4. Antioxidant

The antioxidant activity of the polysaccharides showed that the scavenging ability of the substrate on ABTS radicals and hydroxyl radicals was more similar to that of Vc and was the strongest, followed by mycelia. The scavenging ability of 50% of the alcohol-deposited polysaccharides was significantly stronger than that of the 70% and 90% alcohol-deposited polysaccharides and similar to that of the natural antioxidant vitamin BHT [82].

### 3.5. Others

A previous report revealed that 4,6,8(14),22(23)-tetraene-3-one-ergosterol showed significant antibacterial activity against *Staphylococcus aureus* and *Escherichia coli* [83].

Subsequent studies have shown that *O. lapidescens* polysaccharides induce disease resistance in crops. The *O. lapidescens* polysaccharide LW-1 has been shown to stimulate the activity of defensive enzymes including peroxidase (POD), superoxide dismutase (SOD), and phenylalanine ammonia-lyase (PAL) in tobacco leaves. This is accompanied by the overexpression of the NPR1, PR1, and PR5 resistance genes as well as an elevated level of salicylic acid (SA). Transcriptomic sequencing revealed that LW-1 activated the MAPK signaling pathway, the plant–pathogen interaction pathway, and the glucosinolate metabolism pathway in Arabidopsis thaliana, thereby inducing disease resistance [84].

The oxidases extracted from *O. lapidescens* have been demonstrated to possess the ability to degrade indole-3-acetic acid, a property that may have potential applications in industrial processes [79]. Furthermore, a lectin isolated from *O. lapidescens* with a molecular weight of 12 kDa has been demonstrated to induce agglutination in human and rabbit blood cells, with galactose inducing a conformational change in the lectin [21]. The content of lectin is also suggested to be included in the quality standards of *O. lapidescens* [85].

## 4. Conclusions and Prospects

The mechanisms underlying the anthelmintic activity of *O. lapidescens* proteases are still unclear, particularly their stability and substrate specificity in the digestive environment. In addition, the genes encoding these proteases have not been identified, limiting their potential for therapeutic and industrial applications. Although in vitro studies have shown that their insecticidal activity is comparable to that of typical synthetic chemicals, the yield and price are so low in comparison that *O. lapidescens* is not usually used as an antiparasitic agent in modern clinical therapy, resulting in a lack of clinical data.

In the field of cancer research, significant progress has been made in the study of the *O. lapidescens* pPeOp protein, revealing its role in regulating cancer cell processes and providing a theoretical basis for clinical applications. Traditional Chinese medicine associates *O. lapidescens* with the gastrointestinal meridians, and modern scientific research has validated its inhibitory effects on gastric cancer cells. This integration of traditional and modern approaches underscores its potential in cancer treatment. However, the properties of pPeOp proteins, such as amino acid sequences and encoding genes, are still not fully understood, limiting their further development and use.

The molecular weight suggests that the pPeOP protein and the mercapto protease in *O. lapidescens* protease are the same protein. Therefore, the pPeOp protein faces similar problems to the previously mentioned proteases. pPeOp proteins have great potential for development in the field of biopharmaceuticals. However, the lack of basic research data, in particular detailed analysis of amino acid composition, structure and substrate specificity, has limited their further development and market application. In order to commercialize pPeOp proteins, systematic basic research must first be carried out, not only to determine their molecular structures, but also to elucidate their pharmacological activities under different pathological conditions. In parallel, related biopharmaceutical process technologies including production, purification, and formulation development must be developed to ensure clinical and commercial viability. These are also the parts that need to be improved in the research and industrialization of *O. lapidescens* protease.

Natural polysaccharides have been demonstrated to play an important role in the treatment of disease, particularly in the context of anti-tumor and immunomodulatory functions. Nevertheless, research on *O. lapidescens* polysaccharides has almost come to a standstill over the past few decades. On the one hand, the bioactivities of *O. lapidescens* polysaccharides are plentiful but unremarkable. Their antitumor, anti-inflammatory, antioxidant, and anti-tobacco mosaic virus activities do not show advantages over similar products with the same functions. On the other hand, the insufficient production and high price of *O. lapidescens*, coupled with the insufficient research on the mechanism of polysaccharide bioactivity, present significant challenges to the development of *O. lapidescens* polysaccharide nutritional products that draw on the commercialization of Ganoderma lucidum or other edible mushrooms of a similar type.

Research on *O. lapidescens* triterpenoids is relatively limited, primarily focusing on chemical structure characterization, while their biological functions and mechanisms remain largely unexplored. This presents a broad scope for future studies, especially regarding their potential biological activities.

Despite some progress in *O. lapidescens* bioinformatics research, its genomic data remains incomplete, hindering more in-depth studies in transcriptomics, proteomics, and metabolomics. Completing its genomic data will help facilitate the heterologous expression of its metabolites and advance both basic and applied research.

With the rapid development of the edible and medicinal fungi industry, the artificial cultivation of *O. lapidescens* also shows promise. However, current market demand has not been fully met, particularly due to the lack of scientific guidance on cultivation techniques. Shortening cultivation cycles, increasing sclerotia yield, and optimizing cultivation models are the keys to achieving efficient production. Addressing these issues will greatly enhance the commercialization of *O. lapidescens*.

*O. lapidescens*, as a medicinal fungus, has a long history of use. Its low toxicity and favorable safety profile have made it widely applied in research on various therapeutic effects. Its pharmacological components and metabolites exhibit a wide range of biological activities, indicating great potential for applications in modern medicine. Further research into these components and their mechanisms of action will help deepen the understanding of its therapeutic value, promote the development of new drugs, and enhance its commercial value.

To promote the application of *O. lapidescens* in industry, it is crucial to accelerate research into its biological characteristics and pharmacological mechanisms, complete its genomic information, and optimize cultivation techniques. These efforts will not only expand its industrial and medical applications, but also unlock its potential economic and therapeutic value. Future research should cover both fundamental scientific questions in the laboratory and address practical production needs, thereby advancing its commercialization and medical use.

## Figures and Tables

**Figure 1 ijms-25-11016-f001:**
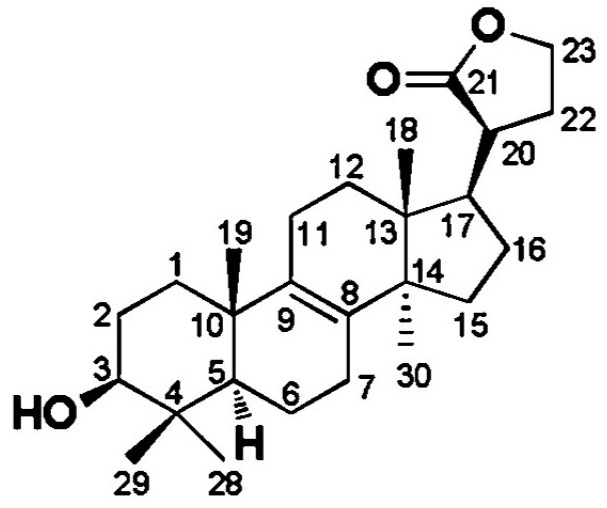
Structural formula of the novel tetranorlanostane triterpenoid isolated from *O. lapidescens*.

**Figure 2 ijms-25-11016-f002:**
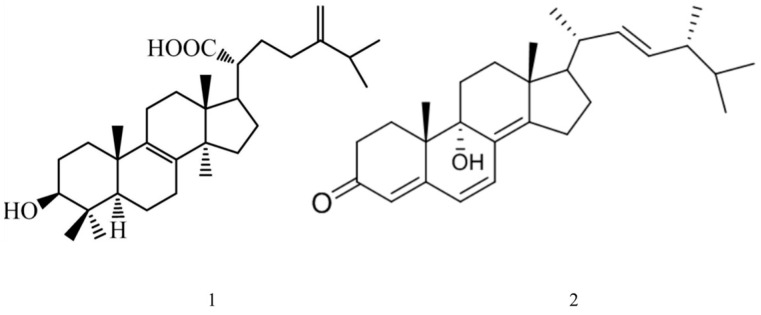
Structural formulae of eburicoic acid(**1**) and ganoderma side D(**2**).

**Figure 3 ijms-25-11016-f003:**
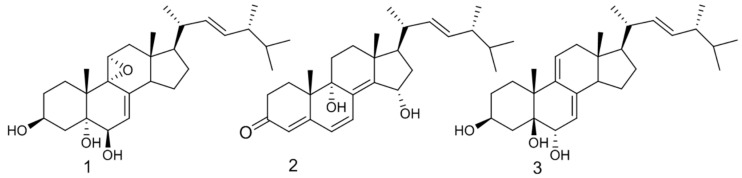
Structural formulae of new ergosterols isolated from *O. lapidescens*: (22E,24R)-9α,11α-epoxyergosta-7,22-diene-3β,5α,6α-triol (Compound **1**), (22E,24R)-9α,15α-dihydroxyergosta-4,6,8(14),22-tetraene-3-one (Compound **2**), and (22E,24R)-ergosta-7,9(11),22-triene-3β,5β,6α-triol (Compound **3**).

**Figure 4 ijms-25-11016-f004:**
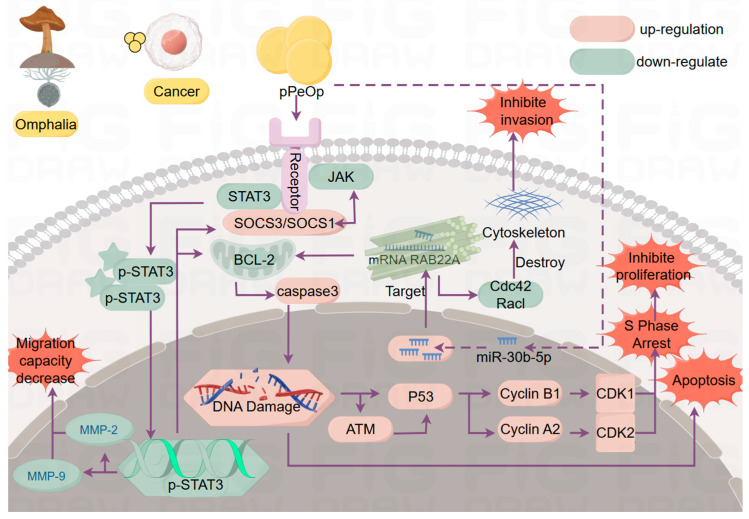
Mechanism of the antitumor activity of *O. lapidescens* protein pPeOp (drawn with Figdraw).

**Figure 5 ijms-25-11016-f005:**
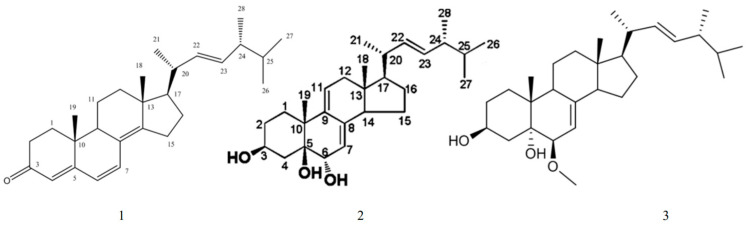
Structural formulae of ergosterol: ergosta-4,6,8(14),22-tetraen-3-one (Compound **1**); (22E,24R)-ergosta-7,9(11),22-trien-3β,5β,6α-triol (Compound **2**); (3β,5α,6β,22E)-6-methoxyergosta-7,22-dien-3,5-diol (Compound **3**).

**Table 1 ijms-25-11016-t001:** Categories of ergosterols isolated from *O. lapidescens* and their biological activities.

Classify	Compound Name	Biological Activities
Epoxides	5α,6α-epoxyergosta-8,22-diene-3β,7α-diol	Indicated antibacterial activities against Escherichia coli and showed potent inhibitory activity against the proliferation of CACO-2, WiDr, DLD-1, and Colo320 human colorectal adenocarcinoma cells [44].
	(22E,24R)-9α,11α-epoxyergosta-7,22-diene-3β,5α,6α-triol	First discovered from *O. lapidescens* (Figure 3, Compound **1**).
	5α,6α-epoxy-3β-hydroxyergosta-22-ene-7-one	No bioactivity was observed.
	5α,6α-epoxy-3β-hydroxy-(22E)-ergosta-8(14),22-dien-7-one	No bioactivity was observed.
	22E-3β-hydroxy-5α,6α,8α,14α-diepoxyergosta-22-en-7-one	Isolated from *Aspergillus awamori*; mildly toxic to A549 [45].
	5α,6α-epoxyergosta-8(14),22-diene-3β,7α-diol	Isolated from *Pleurotus eryngii*, inhibits aromatase [46].
Hydroxyketones	(22E,24R)-9α,15α-dihydroxyergosta-4,6,8(14),22-tetraene-3-one	First discovered from *O. lapidescens* (Figure 3, Compound **2**).
	5α,6α-dihydroxydihydroergosterol	No bioactivity was observed.
	3β,5α-dihydroxy-(22E,24R)-ergosta-7,22-dien-6-one	Exhibited strong or moderate cytotoxic activities against MCF-7, A549, Hela, and KB cell [47].
	3β, 5α, 9α-trihydroxy-(22E,24R)-ergosta-7,22-dien-6-one	At a concentration of 100 μg/mL, it exhibits an inhibition rate of 51.1% against human chronic myelogenous leukemia (CML) cell K562 and also shows certain inhibitory effects on other human tumor cells such as HL-60, BGC-823, and HeLa cells [47].
	5,6β-dihydroxy-5α-ergosta-7,22-dien-3β-ylacetate	No bioactivity was observed.
Endoperoxides	(22E)-5α,8α-epidioxyergosta-6,22-dien-3β-ol	Exhibited moderate cytotoxic activity against human prostate cancer cell line LNCaP-C4-2B [48].
		Showed significant anti-rheumatoid arthritis activities, displaying inhibitory effects on the proliferation of MH7 A synovial fibroblast cells [49].
		Inhibited iNOS activity in LPS-induced macrophages and decreased nitrite levels [50].
	(22E)-ergosta-6,9,22-triene-3β,5α,8α-triol	Effectively inhibited estrogen biosynthesis and reduced the mRNA and protein expression levels of aromatase in human ovarian granulosa-like KGN cells [51].
Ergosterol	ergosta-4,6,8(14),22-tetraen-3-one	More common in fungi, serves as novel AchE inhibitor, and prevents early renal injury in aristolochic acid-induced nephropathy rats [52].
	(22E,24R)-ergosta-8,22-diene-3β,5α,6β,7α-tetrol	No bioactivity was observed.
	(22E,24R)-ergosta-7,22-dien-3β,5α,6β-triol (cerevisterol)	From the mycelia of *Lentinus polychrous*, a Thai local edible mushroom. Can inhibit estrogen-induced proliferation of breast cancer T47D cells [53].
	(22E)-ergosta-7,22-diene-3β,5β,6α-triol	No bioactivity was observed.
	(22E,24R)-ergosta-7,22-diene-3β,5α,6α,9α-tetrol	No bioactivity was observed.
	(22E,24R)-3β,5α-dihydroxy-6β-ethoxyergosta-7,22-diene (fomentarol C)	Fomes fomentarius, which has shown slight cytotoxic effects against HCT116 colon cancer cells [54].
	(3β,5α,6β,22E)-6-methoxyergosta-7,22-diene-3,5-diol	No bioactivity was observed.
	(22E,24R)-ergosta-7,9(11),22-triene-3β,5β,6α-trio	First discovered from *O. lapidescens* (Figure 3, Compound **3**).

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
