# Peer review of "Research on the Action and Mechanism of Pharmacological Components of Omphalia lapidescens"

_ijms, 2024, doi:10.3390/ijms252011016_

Round 1
Reviewer 1 Report
Comments and Suggestions for Authors
The review titled “Research on the action and mechanism of pharmacological components of Omphalia lapidescens” provides a preliminary overview of the compounds found in Omphalia lapidescens that exhibit various biological properties. However, its presentation is rather chaotic.
I suggest first commenting on the compounds described thus far, such as:
- Polysaccharides
- Phenolic Compounds
- Terpenoids
- Sterols
- Fatty Acids
- Alkaloids
Additionally, drawing the structures would help to visualise the structure-activity relationship of each compound.
Next, a section should be dedicated to the reported biological properties, indicating sub-sections for properties such as:
- Immunomodulatory Effects
- Antioxidant Activity
- Antimicrobial Activity
- Anticancer Activity
- Antidiabetic Effects
- Anti-inflammatory Properties
Subsequently, each of these should describe the mechanisms involved for the various compounds presented by this plant species.
Once the authors have included these points, the review can be assessed more effectively.
Author Response
Thank you very much for taking the time to review this manuscript. Please find the detailed responses below and the corresponding revision in the re-submitted files. It is notable that the restructuring of the article has resulted in a substantial alteration to the paragraph order in comparison to the previous version. To avoid the introduction of excessive indications of revisions, the additions have been highlighted in yellow, while the colored text has been either rewritten or deleted.
Comments 1: Structure of the manuscript
Respone1: In accordance with the recommendations, the pharmacologic and bioactive components have been divided into two parts for description. In this case, the secondary headings under pharmacological constituents are divided into five broad categories according to the following: proteins, polysaccharides, terpenoids, sterols, and others. These are further subdivided into tertiary headings to elucidate the properties of the specific constituents. The outline is as follows (seen in Section 2 “Pharmacological components of O. lapidescens”):
2.1 Protein: 2.1.1 protease、2.1.2 lectin、1.1.3 pPeOp
2.2 Polysaccharide: 2.2.1S4001&S4002,2.2.2 OL-1,OL-2&OL-3
2.3 Triterpenes:2.3.1 Tetranorlanostane triterpenoid,2.3.2 Eburicoic acid &ganoderma side D 2.4 Ergosterol
2.5 Other components
Comments 2: Lack of pictures of compounds
Respone 2: In the manuscript, the sections of the review pertaining to ergosterol and triterpenes addressed the issues of cytotoxicity and compound structure. To facilitate a more comprehensive understanding of these matters, the structural figures of the relevant compounds (drawing with the online website Figdraw) have been incorporated into these sections (seen Figure 1, 2, 3, and 5).
Comments 3: Separate section on biological activity
Respone 3: According the recommendations, the biological activity section was subdivided into secondary headings and listed, with tertiary headings under the antitumor (seen in Section 3 “Biological activity”).
3.1 Antiparasitic
3.2 Antitumor: 3.2.1 Omphalia protein pPeOp, 3.2.2 Omphalia polysaccharides OL-2, 3.2.3 Omphalia triterpene, 3.2.4 Omphalia ergosterol
3.3 Anti-inflammatory
3.4 Antioxidant
3.5 Others
Comments 4: Lack of a specific mechanism for biological activity
Respone 4: Combined with the existing research results, we have enumerated the pertinent pharmacological constituents of diverse biological activities and have categorized their mechanisms. In the section on antitumor activity, the antitumor effects and mechanisms of action of the different compounds are described, and the potential mechanism of pPeOp in regulating the JAK-STAT pathway is added (seen in Section 3.2 “Antitumor”). Nevertheless, there is a paucity of antiparasitic studies on Omphalia, with only a handful conducted in the 2000s. The antiparasitic properties of Omphalia remain to be elucidated in detail at the molecular level. In addition, the functions of many compounds, such as antioxidant and anti-inflammatory properties, have not been fully elucidated. In conclusion, the primary reason for the absence of a comprehensive mechanistic discourse is that the studies were conducted at an early stage and lacked sufficient depth, resulting in a relatively simplistic understanding of the mechanism.
Reviewer 2 Report
Comments and Suggestions for Authors
This review provides a comprehensive summary of the pharmacological components and mechanisms of Omphalia lapidescens (Lei Wan), focusing on its antiparasitic, anticancer, anti-inflammatory, and immunomodulatory properties. Overall, the article has several strengths, but there are areas where improvement and expansion would enhance its clarity, depth, and academic rigor.
Strengths:
-
Comprehensive coverage: The article discusses various active compounds of Omphalia lapidescens (proteases, polysaccharides, sterols, and triterpenoids) and explores their diverse pharmacological effects. This broad approach offers readers a solid understanding of the herb’s medicinal potential.
-
Strong scientific foundation: The review incorporates numerous studies to substantiate Omphalia lapidescens's biological activities, particularly in anticancer and antiparasitic contexts, providing credible support for its claims.
-
Connection between traditional use and modern research: By integrating traditional Chinese medicine knowledge with contemporary scientific findings, the article showcases the continuing relevance of herbal medicine in modern pharmacology, which adds an interesting and valuable perspective.
Areas for Improvement and Expansion:
-
Clearer structure: While the review covers a wide range of topics, some sections could benefit from clearer organization. Currently, the text can feel dense. Adding summaries or tables after each section to highlight key points would improve readability. Additionally, dividing sections more explicitly by mechanism (e.g., antiparasitic, anticancer) would help readers follow the flow of information more easily.
-
Insufficient exploration of mechanisms:
- Although the article highlights the pharmacological effects of key compounds (proteases, polysaccharides, sterols), it lacks detailed discussion of their molecular mechanisms. Providing more information on specific molecular targets or pathways (such as how these compounds act on signaling pathways or cellular processes) would increase the scientific depth of the review.
- For example, while the proteases' antiparasitic effects are mentioned, there is little explanation at the molecular level about how these proteases interact with parasite structures or cellular processes.
-
Comparison with modern treatments:
- The review discusses the therapeutic potential of Omphalia lapidescens but does not compare its effects to current treatments or other natural compounds in similar applications (e.g., chemotherapy drugs or other antiparasitic agents). Including a comparison of its advantages and limitations versus modern drugs could highlight its unique value in medical applications and provide insights into where it fits in the broader therapeutic landscape.
-
Limited forward-looking perspective:
- While the review summarizes existing research well, the section on future research directions is somewhat general. Offering more specific guidance, such as identifying critical unanswered questions (e.g., how to optimize extraction methods, key genes involved, or pharmacokinetics) or pointing out areas for deeper investigation (genomics, metabolomics, or clinical applications), would add value to researchers looking for next steps.
-
Lack of visual aids:
- The review relies heavily on text and references but would benefit from the addition of diagrams, tables, or flowcharts to explain complex processes and relationships. For example, including a figure summarizing the different pharmacological effects of Omphalia’s components or a diagram showing how the compounds interact with specific pathways would make the content more accessible and visually engaging.
Conclusion:
Overall, this review covers a significant amount of important material on Omphalia lapidescens, making it a valuable resource. However, it could be improved by delving deeper into the molecular mechanisms, comparing the herb’s effects with modern treatments, providing clearer direction for future research, and utilizing visual elements to clarify and enhance key points. These additions would strengthen the article’s clarity, depth, and overall impact in the scientific community.
Author Response
Thank you very much for taking the time to review this manuscript. Please find the detailed responses below and the corresponding revision in the re-submitted files. It is notable that the restructuring of the article has resulted in a substantial alteration to the paragraph order in comparison to the previous version. To avoid the introduction of excessive indications of revisions, the additions have been highlighted in yellow, while the colored text has been either rewritten or deleted.
Comments 1: Structure of the manuscript
Respone 1: In accordance with the recommendations, the pharmacologic and bioactive components have been divided into two parts for description. In this case, the secondary headings under pharmacological constituents are divided into five broad categories according to the following: proteins, polysaccharides, terpenoids, sterols, and others. And the biological activity section was subdivided into antiparasitic, antitumor, anti-inflammatory, antioxidant and others.
These are further subdivided into tertiary headings to elucidate the properties of the specific constituents and their mechanisms, respectively. The outline is as follows (seen in Section 2 “Pharmacological components of O. lapidescens” and Section 3 “Biological activity”):
2.1 Protein: 2.1.1 protease、2.1.2 lectin、1.1.3 pPeOp
2.2 Polysaccharide: 2.2.1S4001&S4002,2.2.2 OL-1,OL-2&OL-3
2.3 Triterpenes:2.3.1 Tetranorlanostane triterpenoid,2.3.2 Eburicoic acid &ganoderma side D 2.4 Ergosterol
2.5 Other components
3.1 Antiparasitic
3.2 Antitumor: 3.2.1 Omphalia protein pPeOp, 3.2.2 Omphalia polysaccharides OL-2, 3.2.3 Omphalia triterpene, 3.2.4 Omphalia ergosterol
3.3 Anti-inflammatory
3.4 Antioxidant
3.5 Others
Comments 2: Insufficient exploration and discussion of mechanisms
Respone 2: Combined with the existing research results, we have enumerated the pertinent pharmacological constituents of diverse biological activities and have categorized their mechanisms. In the section on antitumor activity, the antitumor effects and mechanisms of action of the different compounds are described, and the potential mechanism of pPeOp in regulating the JAK-STAT pathway is added (seen in Section 3.2 “Antitumor”). Nevertheless, there is a paucity of antiparasitic studies on Omphalia, with only a handful conducted in the 2000s. The antiparasitic properties of Omphalia remain to be elucidated in detail at the molecular level. In addition, the functions of many compounds, such as antioxidant and anti-inflammatory properties, have not been fully elucidated. In conclusion, the primary reason for the absence of a comprehensive mechanistic discourse is that the studies were conducted at an early stage and lacked sufficient depth, resulting in a relatively simplistic understanding of the mechanism.
Comments 3: Lack of comparison of modern therapies
Respone 3: Despite its status as an anthelmintic herbal medicine, Omphalia is unable to compete with modern synthetic drugs in terms of clinical use due to yield and cost limitations. This is also a consequence of the difficulties encountered in the cultivation of Omphalia and the extraction of its pharmacological components. Consequently, there is a paucity of comparative clinical data on parasite treatment. With regard to antitumor activity, Omphalia tablets and Omphalia capsules, among other formulations, have undergone the requisite clinical validation as adjuvant cancer therapeutic drugs. This is also indicated in the original article in the Omphalia protein pPeOp section. Based on this, we have made some additions and listed the specific data, which can be seen in Section 3.2 “Antitumor” with Paragraph two.
Comments 4: Limited forward-looking perspective
Respone 4: As previously stated, significant issues persist regarding the research on Omphalia. The scientific designation of Omphalia remains a subject of contention. Furthermore, there is a significant deficit in genetic data pertaining to Omphalia. At present, only the ITS sequence and an unvalidated mitochondrial sequence of Omphalia are available on NCBI. Information on its bioactive proteins, metabolic gene clusters, and other relevant data remain absent. Consequently, Omphalia 's prospective study is constrained by these limitations. In conclusion, future research on Omphalia should prioritize the supplementation and determination of genetic information, complemented by research on cultivation and extraction processes (seen in Section 4 “Conclusion and Prospects” with Paragraphs three, four, five).
Comments 5: Lack of visual aids
Respone 5: We have supplemented the compound structure diagrams (drawing with the online website Figdraw) in the triterpenoids and ergosterol chapters (seen Figure 1, 2, 3, and 5). Furthermore, the antitumor mechanism of Omphalia pPeOp protein was redrawn with greater tension in the diagram (seen Figure 4).
Round 2
Reviewer 1 Report
Comments and Suggestions for Authors
The authors of the review titled "Research on the action and mechanism of pharmacological components of Omphalia lapidescens" have made the revisions suggested by the reviewers. The manuscript has significantly improved and can now be published on the journal's platform.
Reviewer 2 Report
Comments and Suggestions for Authors
The authors have addressed all my concerns, and I recommend accepting the manuscript.